



# Varying regional $\delta^{18}$O–temperature relationship in high resolution stable water isotopes from East Greenland

Christian Holme[a], Vasileios Gkinis[a], Mika Lanzky[a,b], Valerie Morris[c], Martin Olesen[d], Abigail Thayer[c], Bruce H. Vaughn[c], and Bo M. Vinther[a]

[a]Centre for Ice and Climate, The Niels Bohr Institute, University of Copenhagen, Denmark
[b]Department of Geosciences, University of Oslo, Norway
[c]Institute of Arctic and Alpine Research, University of Colorado Boulder, Boulder, Colorado, USA
[d]Danish Climate Centre, Danish Meteorological Institute, Copenhagen, Denmark

**Correspondence:** Christian Holme (christian.holme@nbi.ku.dk)

**Abstract.** This study examines the stable water isotope signal ($\delta^{18}$O) of three ice cores drilled on the Renland peninsula (East Greenland coast). While ice core $\delta^{18}$O measurements qualitatively are a measure of the local temperature history, the $\delta^{18}$O variability actually reflects the integrated hydrological activity that the deposited ice experienced from the evaporation source to the condensation site. Thus, as Renland is located next to a fluctuating sea ice cover, the transfer function used to infer past temperatures from the $\delta^{18}$O variability is potentially influenced by variations in the local moisture conditions. The objective of this study is therefore to evaluate the $\delta^{18}$O variability of ice cores drilled on Renland and examine what amount that can be attributed to regional temperature variations. In the analysis, three ice cores are utilized to create stacked summer, winter and annually averaged $\delta^{18}$O signals (AD 1801-2014). The imprint of temperature on $\delta^{18}$O is first examined by correlating the $\delta^{18}$O stacks with instrumental temperature records from East Greenland (AD 1895-2014) and Iceland (AD 1830-2014) and with the regional climate model HIRHAM5 (AD 1980-2014). The results show that the $\delta^{18}$O variability correlates with regional temperatures on both a seasonal and an annual scale between 1910-2014 while $\delta^{18}$O is uncorrelated with Iceland temperatures between 1830-1909. Our analysis indicates that the unstable regional $\delta^{18}$O-temperature correlation does not result from changes in weather patterns through respectively strengthening and weakening of the North Atlantic Oscillation. Instead, the results imply that the varying $\delta^{18}$O-temperature relation is connected with the volume flux of sea ice exported through Fram Strait (and south along the coast of East Greenland). Notably, the $\delta^{18}$O variability only reflects the variations in regional temperature when the temperature anomaly is positive and the sea ice export anomaly is negative. It is hypothesized that this could be caused by a larger sea ice volume flux during cold years which suppresses the Iceland temperature signature in the Renland $\delta^{18}$O signal. However, more isotope-enabled modeling studies with emphasis on coastal ice caps are needed in order to quantify the mechanisms behind this observation. As the amount of Renland $\delta^{18}$O variability that reflects regional temperature varies with time, the results have implications for studies performing regression-based $\delta^{18}$O-temperature reconstructions based on ice cores drilled in the vicinity of a fluctuating sea ice cover.



## 1 Introduction

Polar ice caps store deposited precipitation as stratified ice layers thousands of years back in time. This precipitation consists of stable water isotopes ($\delta^{18}$O, $\delta$D) that work as a direct proxy of the relative depletion of a water vapor mass in its transport from the evaporation source to the site where condensation takes place (Epstein et al., 1951; Mook, 2000). This traceability manifests as a correlation between $\delta^{18}$O and the temperature in the cloud at the time of condensation (Dansgaard, 1954, 1964). Thus, a relationship between $\delta^{18}$O and temperature is preserved in annual layers of precipitation on an ice cap. Hence, by drilling ice cores at polar sites such as Antarctica and Greenland, it is possible to access past temperatures imprinted in the $\delta^{18}$O signal. Several studies have examined the relation between temperature and ice core $\delta^{18}$O, and its linear or quadratic relationship has regularly been used as a transfer function to infer past temperature (Jouzel and Merlivat, 1984; Johnsen et al., 1989; Jouzel et al., 1997; Johnsen et al., 2001; Ekaykin et al., 2017). While it is evident that $\delta^{18}$O and temperature covary, the $\delta^{18}$O signal is also affected by changes in sea ice and atmospheric circulation (Noone and Simmonds, 2004; Steig et al., 2013). Changes in the atmospheric circulation can be triggered by climatic oscillation modes (e.g. the North Atlantic Oscillation, Southern Annual Mode, Pacific Decadal Oscillation etc.) which affect the $\delta^{18}$O variability as they influence precipitation patterns (Vinther et al., 2010; Ekaykin et al., 2017). Additionally, changes in sea ice extent affect the local moisture conditions which particularly influence the coastal precipitated $\delta^{18}$O variability (Bromwich and Weaver, 1983; Noone and Simmonds, 2004). Such variations have implications for a simple regression-based reconstruction of temperature from $\delta^{18}$O as the variability patterns between the ice core isotope signal and the oscillation modes and sea ice extent can have varied in strength back in time. Furthermore, in studies that analyze the relationship between polar precipitated $\delta^{18}$O and temperature, the temperature record is often substantially shorter than the $\delta^{18}$O series. While this is inevitable when performing temperature reconstructions, utilizing a short temperature record complicates the possibility of verifying if the estimated $\delta^{18}$O-temperature relation is stable with time.

The aim of this study is to examine how much of the $\delta^{18}$O variability (AD 1801-2014) from a stack of ice cores drilled on the Renland peninsula, Eastern Greenland, can be attributed to temperature variations (map in Fig. 1). In the analysis, seasonally averaged $\delta^{18}$O time series have been compared with regional temperatures through instrumental temperature records located on the coasts of East Greenland (AD 1895-2014) and Iceland (AD 1821-2014) and the regional atmospheric climate model HIRHAM5 $2\,\mathrm{m}$ temperature output (AD 1980-2014). The $\delta^{18}$O signal is divided into its seasonal components as it potentially improves the reconstruction of variability in weather regimes and past temperatures (Vinther et al., 2003b, 2010; Zheng et al., 2018). As Renland is located at the coast, its hydrological conditions are connected with the sea ice that annually is transported south along the coast of East Greenland. Relatively small loss in regional sea ice extent ($\approx 10\%$ or less) has previously been found to influence local Greenland moisture source water vapor which is traceable in the corresponding ice core deuterium excess values (Klein and Welker, 2016). The deuterium excess signal ($\mathrm{d_{xs}} = \delta\mathrm{D} - 8 \cdot \delta^{18}\mathrm{O}$, (Dansgaard, 1964)) contains information about the kinetic fractionation occurring when moisture evaporates from the ocean surface and ice core $\mathrm{d_{xs}}$ has been found to correlate with relative humidity and sea surface temperature at the source region (Jouzel and Merlivat, 1984; Johnsen et al., 1989). Thus, besides investigating the regional $\delta^{18}$O–temperature relationship, this study examines if changes in the Arctic sea ice extent can be detected in the Renland stable water isotopes.



## 2  The ice cores

The Renland ice cap has an area of $1200\,\mathrm{km}^2$ with an average ice thickness of a few hundred meters. It is separated from the main Greenlandic ice sheet as a small peninsula on the east coast of Greenland (map in Fig. 1). The ice cap experiences a high annual accumulation rate of around $0.47\,\mathrm{m\,ice/year}$ with an annual surface temperature of $-18^{\circ}\mathrm{C}$. Renland has probably never

been overridden by the Inland ice as it is surrounded by deep branches of the Scoresbysund Fjord which effectively drains the Inland ice (Johnsen et al., 1992). Additionally, the small width of the ice cap which is constrained by the surrounding mountains prohibits the ice elevation from significant increases from present day height. As a result, the ice cap has not experienced any ice sheet elevation changes for the past 8000 years (except for slight uplift due to isostatic rebound) (Vinther et al., 2009). This implies that lapse-rate controlled temperature variations resulting from a varying ice sheet thickness will be negligible.

This study utilizes three ice cores drilled on Renland in the analyses (Table 1). Two cores were drilled next to each other in the year 1988 (main (M) and shallow (S) cores) while the third was drilled approximately $2\,\mathrm{km}$ away in 2015 as part of the REnland ice CAP project (RECAP). The 1988 M and RECAP cores extend over the past 120 ka while the 1988 S core only covers the time back to the year 1801. Despite two cores covering the past interglacial and glacial period, the study focuses on the period AD 1801-2014 as we here have three overlapping ice core records and instrumental temperature recordings.

Moreover, the separation of the summer and winter signals is better facilitated when the annual layers not are obliterated due to diffusion and ice thinning.

The records from 1988 were measured on a Isotope Ratio Mass Spectroscopy (IRMS) with a discrete resolution of $5.0\,\mathrm{cm}$ while the RECAP core was measured using cavity ringdown spectroscopy (Picarro L2130) on a Continuous Flow Analysis (CFA) system with a nominal resolution of $0.5\,\mathrm{cm}$. For the RECAP core, the years 2011-2014 are covered by the snow pit core

A6 drilled next to the drill site. The A6 core was measured discretely with a sample size of $5.0\,\mathrm{cm}$ on a Picarro L2130. This was done as the porous snow in the upper firn column easily inhibits a stable measurement flow in the CFA analysis which complicates precise water isotope measurements.

**Table 1.** The subset of the three ice cores used in this study: processing information, analysis and coordinates.

| Cores | Coordinates | Time span | Depth span | Meas. | Resolution | Analysis |
|-------|-------------|-----------|------------|-------|------------|----------|
| RECAP | $71^{\circ}\,18'\,18''\,\mathrm{N}; 26^{\circ}\,43'\,24''\,\mathrm{W}; 2315\,\mathrm{m.a.s.l.}$ | $1801-2014$ | $111.7\,\mathrm{m}$ | $\delta^{18}\mathrm{O}, \delta\mathrm{D}$ | $0.5\,\mathrm{cm}$ | CFA-L2130 |
| 1988 M | $71^{\circ}\,18'\,17''\,\mathrm{N}; 26^{\circ}\,43'\,24''\,\mathrm{W}; 2340\,\mathrm{m.a.s.l.}$ | $1801-1987$ | $92.5\,\mathrm{m}$ | $\delta^{18}\mathrm{O}$ | $5.0\,\mathrm{cm}$ | IRMS |
| 1988 S | $71^{\circ}\,18'\,17''\,\mathrm{N}; 26^{\circ}\,43'\,24''\,\mathrm{W}; 2340\,\mathrm{m.a.s.l.}$ | $1801-1987$ | $91.6\,\mathrm{m}$ | $\delta^{18}\mathrm{O}$ | $5.0\,\mathrm{cm}$ | IRMS |

## 3  Diffusion correction

Firn diffusion dampens the annual oscillations in the $\delta^{18}\mathrm{O}$ data. This takes place while firn (snow that survived a season) is

transformed into ice in the top 60-80 meters of the ice sheet. During this densification process, air in the open porous firn

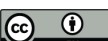



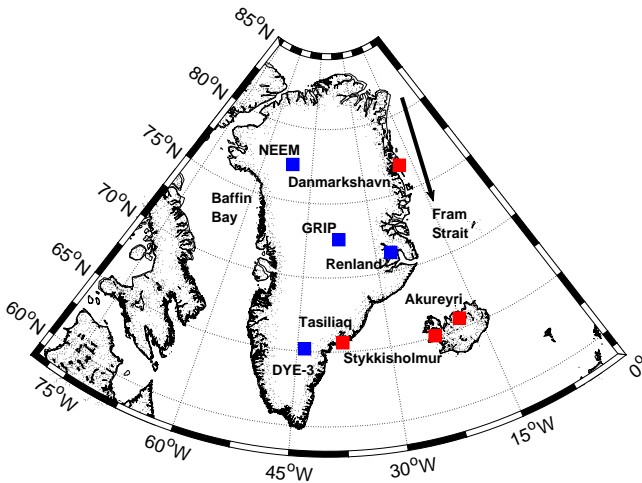

**Figure 1.** Locations of ice core drill sites (blue squares) and the instrumental temperature records (red squares). Sea ice exported from Fram Strait is represented by the black arrow.

is interconnnected which enables the stable water isotopes in the firn air to mix with the snow grains (Johnsen, 1977). This molecular diffusion process makes the $\delta^{18}$O signal become increasingly more smooth with depth until pore close-off. The firn diffusion of stable water isotopes imposes two challenges on the analysis presented in this study. First, the diffusion of the annual oscillations creates artificial trends in summer and winter season time series of $\delta^{18}$O (Vinther et al., 2010). Secondly, it introduces a bias when comparing the ice cores drilled in 1988 with the ice core drilled in 2015. For instance, the $\delta^{18}$O signal representing the year 1987 has only experienced 1 year of firn diffusion in the 1988 ice cores while it has experienced 28 years of firn diffusion in the 2015 core. The $\delta^{18}$O signal for overlapping years will therefore be more attenuated in the 2015 core.

As this study compares the seasonally averaged $\delta^{18}$O signals of three ice cores drilled in different years, it is necessary to ensure that each $\delta^{18}$O record has the same firn diffusive properties with depth. This is typically achieved by correcting each $\delta^{18}$O record such that the effect of increasing smoothing with depth is removed by deconvolving the measured $\delta^{18}$O signal to restore the originally deposited signal. However, Renland frequently experiences summer melting which causes steep isotopic gradients in the firn. Such high frequency gradients complicate a deconvolution of the measured $\delta^{18}$O signal (Cuffey and Steig, 1998; Vinther et al., 2010). Instead, this study forward diffuses the three $\delta^{18}$O records with depth such that each $\delta^{18}$O series has been influenced by the same amount of firn diffusion. Diffusion of stable water isotopes is typically described by the diffusion length ($\sigma$) which is the average vertical displacement of a water molecule (units in meters). Thus, the $\delta^{18}$O series are forward diffused ($\delta^{18}$O$_{fd}$) such that each record has the same $\sigma$ with depth. Despite that such a smoothing procedure slightly mixes the summer and winter signals, a distinction of the seasonal components is still possible due to Renland's thick annual layers greatly exceeding the diffusion length.



The procedure below outlines in three steps how this was done separately for the 2015, 1988 M and 1988 S cores.

Step 1: First, the amount of diffusion that the measured $\delta^{18}$O signal already has experienced with depth is computed through the diffusion length's density dependence (for origin see Gkinis et al. (2014); Holme et al. (2018)):

$$\sigma^2(\rho) = \frac{1}{\rho^2} \int_{\rho_o}^{\rho} 2\rho^2 \left(\frac{d\rho}{dt}\right)^{-1} D(\rho)\, d\rho \tag{1}$$

where $D(\rho)$ is the firn diffusivity and $d\rho/dt$ is the densification rate. This study uses the firn diffusivity parameterization of Johnsen et al. (2000) (described in Appendix A) that employs the site–dependent parameters of temperature ($-18\,^{\circ}$C), accumulation rate (0.47 m ice/year), surface pressure (0.75 atm) and density. Density is here modeled with depth by fitting a Herron and Langway (1980) densification model to density measurements from the drill sites. From Eq. 1, it is possible to calculate the diffusion length that each layer has experienced ($\sigma^2(\rho)$) (left subplot in Fig. C1).

Step 2: Equation 1 can be used to calculate the auxiliary diffusion needed to transform a $\delta^{18}$O record into having a uniform diffusion length independent of depth. An auxiliary diffusion ($\sigma^2(\rho)_{aux}$) is calculated as the difference between the final diffusion length at the pore close–off density ($\sigma^2(\rho_{pc} = 804.3\,\text{kg/m}^3)$) and the diffusion length at a given layer in meters of ice–equivalent depth:

$$\sigma^2(\rho)_{aux} = \left( \left(\frac{\rho_{pc}}{\rho_i}\right)^2 \sigma^2(\rho_{pc}) - \left(\frac{\rho(z)}{\rho_i}\right)^2 \sigma^2(\rho) \right) \cdot \left(\frac{\rho_i}{\rho(z)}\right)^2 \tag{2}$$

where the fraction $\rho_i/\rho(z)$ ultimately is multiplied in order to transform the $\sigma^2(\rho)_{aux}$ from representing ice–equivalent depth to density–equivalent depth (as the annual oscillations are squeezed during firn compaction). Using Eq. 2, an auxiliary diffusion profile with respect to density (and thus depth) is calculated for an ice core (left subplot in Fig. C1).

Step 3: Forward diffusion is then simulated through a convolution of the measured data ($\delta^{18}$O$_{meas}$) with a Gaussian filter ($\mathcal{G}$) with a standard deviation equal the auxiliary diffusion length as this is mathematically equivalent to firn diffusion (Johnsen, 20  1977):

$$\delta^{18}\text{O}_{fd}(z) = \delta^{18}\text{O}_{meas} * \mathcal{G} \tag{3}$$

where

$$\mathcal{G}(z) = \frac{1}{\sigma_{aux}\sqrt{2\pi}}\, e^{-z^2/\left(2\sigma_{aux}^2\right)} \tag{4}$$

As the auxiliary diffusion length decreases with depth, the width of the Gaussian filter changes accordingly. Thus, the convo-
lution (using the $\sigma_{aux}$ for the corresponding depth) is applied on a moving $2\,\text{m}$ section which is shifted in small steps equal to the sampling interval. For each convolved data section, only the midpoint of the sliding window is retained as the new forward diffused $\delta^{18}$O$_{fd}$ value. In order to avoid tail-problems when diffusing the top 2 meter measurements, the $\delta^{18}$O$_{meas}$ data were extended by using its prediction filter coefficients estimated from a maximum entropy method algorithm by Andersen (1974). This assumes that the extended series has the same spectral properties as the original series. After applying this smoothing 30  routine on the entire record, a $\delta^{18}$O$_{fd}$ series with constant $\sigma$ is obtained. A comparison between $\delta^{18}$O$_{fd}$ and $\delta^{18}$O$_{meas}$ is shown in Fig. C1.




## 4   Chronology

It is important to ensure that the chronologies of the three ice cores are synchronous before comparing the $\delta^{18}$O variability. The two cores drilled in 1988 were manually dated by counting the summer maxima and winter minima in the $\delta^{18}$O series and verified by identifying signals of volcanic eruptions in the electrical conductivity measurements (Vinther et al., 2003b,

2010). For the 2015 RECAP core, the period 1801-2007 was dated with the annual layer algorithm (StratiCounter) presented in Winstrup et al. (2012) and the years 2007-2014 were manually counted similar to the 1988 cores (the RECAP chronology is presented in Simonsen et al. (2018, in review)). The annual layer algorithm uses signals in the ice core that all have annual oscillations or peaks such as the chemical impurities ($\mathrm{Na}^+$, $\mathrm{Ca}$, $\mathrm{SO}_4^{2-}$ and $\mathrm{NH}_4^+$), electrical conductivity and stable water isotopes. Even though the model automatically count years, the chronology is still restricted by the same volcanic eruptions as

in the 1988 cores. The model marks a year when $\mathrm{Na}^+$ has a peak which indicates winter. $\mathrm{Na}^+$ is a result of the transport of salt from the ocean and it peaks during winter due the strong winds during the fall. As the timing of this winter peak might not be similar to the timing of the $\delta^{18}$O series' winter minima (used for the 1988 cores), this study has tuned the RECAP dating presented Simonsen et al. (2018, in review) slightly. For each year, this is done by tuning the timing of the summer and winter in the dated RECAP record to match with the maximum and minimum of the $\delta^{18}$O series. The chronology is only

shifted a maximum of a few months and it is only changed within a given year. This ensures that the modified dating profile remains consistent with the original chronology while it facilitates an optimal comparison between the manually dated and the automatically dated stable water isotopes profiles.

In order to analyze the seasonal signals of the $\delta^{18}$O series we need to distinguish between snow deposited during summer and winter. Under the assumption that $\delta^{18}$O and temperature extremes in the Greenland region occur simultaneously, Vinther et al.

(2010) found best to define the summer and winter seasons such that they each contain $50\%$ of the annual accumulation. Besides maximizing the amount of utilized data, this definition ensures that the winter and summer signals contain no overlapping data. This study has therefore defined the summer and winter seasons similar to Vinther et al. (2010). The summer, winter and annually averaged $\delta^{18}$O data used in this study are thus seasonal/annual averages of the forward diffused $\delta^{18}$O series.

## 5   $\delta^{18}$O variability on Renland

The three ice cores' $\delta^{18}$O data as a representative of the isotope hydrology on Renland is first evaluated by calculating Pearson correlation coefficients and signal to noise variance ratios (SNR) on the forward diffused $\delta^{18}$O records in the overlapping period 1801-1987. The correlation coefficient is a metric that describes the linear relation between two signals and it has been calculated for different combinations of the presented ice cores (Table 2). For all correlation coefficient calculations throughout this study, the level of significance is estimated based on a Monte Carlo routine described in Appendix B. From the results

displayed in Table 2, it is evident that the lowest correlation coefficients are found for the winter averaged data with values ranging from $0.60 - 0.78$ while the summer and annually averaged signals have higher values ranging from approximately $0.64 - 0.84$. The high correlation coefficients indicate that there is a strong linear relationship between the $\delta^{18}$O records. This is further illustrated by the visual covariation of the annually averaged $\delta^{18}$O records in Fig. 2. In all instances, the highest




correlations are found when correlating the two ice cores drilled in 1988. This might be attributed to the use of similar dating method and their close proximity. Nonetheless, all the presented ice cores correlated significantly during the 1801-1987 period.

**Table 2.** Correlation coefficients ($r$) calculated for different combinations of $\delta^{18}O$ records for the period 1801-1987 ($p < 0.05$).

| Season | $r$ (2015/1988 M) | $r$ (2015/1988 S) | $r$ (1988 M/1988 S) |
|--------|-------------------|-------------------|---------------------|
| Winter | 0.63 | 0.60 | 0.78 |
| Summer | 0.66 | 0.65 | 0.82 |
| Annual | 0.64 | 0.66 | 0.84 |

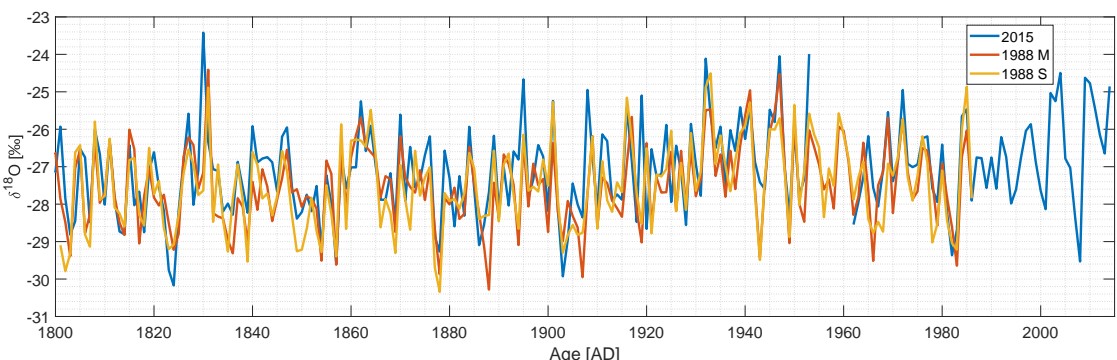

**Figure 2.** Annually averaged $\delta^{18}O$ for respectively the RECAP 2015 (blue), 1988 M (red) and 1988 S (yellow) cores with age.

The $\delta^{18}O$ variability can be further analyzed by examining the mean single series SNR which provides an insight into the amount of signal and noise in the $\delta^{18}O$ series. Noise can originate from depositional effects such as wind shuffling of snow, melt layers and from dating uncertainties ($\pm 1\,\text{year}$) in between the three cores. By averaging $n$ (3) overlapping ice core data records, the mean single series SNR is calculated by comparing the variance of an averaged record ($\text{VAR}_s$) with the mean of the variances ($\overline{\text{VAR}}$) for the $n$ individual records (Johnsen et al., 1997; Vinther et al., 2006):

$$\text{SNR} = \frac{\text{VAR}_s - \frac{\overline{\text{VAR}}}{n}}{\overline{\text{VAR}} - \text{VAR}_s} \tag{5}$$

The SNR results are shown in Table 3. Similar to the high correlation, it is evident that merging the two 1988 records results in the highest SNR values. Moreover, the summer averaged signal has a higher SNR compared to the winter averaged signal which probably is a consequence of winters having more windy conditions that generates redeposition of snow. A similar pattern have previously been found for the seasonal isotopes of GRIP ($n = 5$; SNR summer: 0.70, winter: 0.51), Dye-3 ($n = 2$; SNR summer: 1.73, winter: 1.56) and NEEM ($n = 4$; SNR summer: 1.28, winter: 0.64) (map in Fig. 1) (Vinther, 2003a; Zheng et al., 2018). This comparison also shows that the SNR values of the three Renland ice cores are high compared to GRIP, Dye-3



**Table 3.** Mean signal to noise variance ratios calculated for the summer, winter and annually averaged data using respectively two and three cores in the period 1801-1987.

| Merged cores | SNR winter | SNR summer | SNR annual |
|---|---|---|---|
| 1988 M, 2015 | 1.65 | 1.73 | 1.73 |
| 1988 M, 1988 S | 3.53 | 4.46 | 5.05 |
| 1988 M, 1988 S, 2015 | 2.01 | 2.36 | 2.43 |

and NEEM which likely can be attributed to a combination of a high accumulation rate and a good cross-dating between the compared cores.

From this analysis, the study can comment on two things. First, the two 1988 cores have the most robust common signal of all the tested combinations. As this was for two adjacently drilled ice cores, utilizing all three records still result in a larger spatial atmospheric representativeness of the region. Secondly, the high SNR and correlation coefficients imply that the chronologies from the annual layer detection algorithm and the manual counting are consistent. This has implications for future ice core science as manual layer counting can be a slow and inefficient procedure. Thus, manual counting can effectively be replaced with the StratiCounter software by Winstrup et al. (2012) for ice cores where several datasets that contain observable annual peaks or oscillations are available.

The high combined SNR values and correlation coefficients indicate that it is beneficial to combine the time series into a stacked $\delta^{18}O$ record. We choose to employ all three ice cores as that increases the spatial representativeness of $\delta^{18}O$ while it provides water isotopic variability for the years 1988-2014. A stacked record is typically created by averaging the time series but the time span 1801-2014 consists of an inhomogeneous amount of data records as only the RECAP core contains data in the 1988-2014 period while it also has a gap between 1954-1961 due to missing ice samples. Thus, it is important to implement a variance correction in order to avoid bias issues when averaging time series with nonuniform length (Osborn et al., 1997; Jones et al., 2001). This variance correction ($c$) can be expressed directly through the SNR values in Table 3 and the number of records ($m$) used in the averaging for the given year (derivation can be found in Vinther et al. (2006)):

$$c = \sqrt{\frac{\text{SNR}}{\text{SNR} + \frac{1}{m}}} \tag{6}$$

Before stacking, the three time series are standardized based on the period of overlap (1801-1987) ($\delta^{18}O_{\text{std}}$ has mean = 0 and standard deviation = 1). An average $\delta^{18}O_{\text{avr}}$ value is then calculated by multiplying $c$ onto the mean $\delta^{18}O_{\text{std}}$ for each year:

$$\delta^{18}O_{\text{avr}} = c \cdot \frac{1}{m} \sum_{i=1}^{m} \delta^{18}O_{\text{std}_i} \tag{7}$$

The amplitude and variability of the original $\delta^{18}O$ series are then restored by using the average variance ($\overline{\text{VAR}}$) and the average ($\overline{\delta^{18}O}$) of the three time series (from the period where the time series were standardized):

$$\delta^{18}O_{\text{stack}} = \delta^{18}O_{\text{avr}} \cdot \sqrt{\overline{\text{VAR}}} + \overline{\delta^{18}O} \tag{8}$$




Figure 3 shows the summer, winter and annual $\delta^{18}O_{stack}$ series for the period 1801-2014. In the figure, a 5 year moving average has been applied on the stacked records in order filter out any remaining high frequency noise variability. From the figure, it is evident that the summer averaged signal is less depleted than the annual and winter averaged signals. Moreover, the summer signal has the largest trend in $\delta^{18}O$ with an increase of $0.54‰/\text{century}$ while the winter and annually averaged

data show lower increases of respectively $0.24‰/\text{century}$ and $0.37‰/\text{century}$. The amount of variability that correlates with temperature will be examined in Sect. 6.

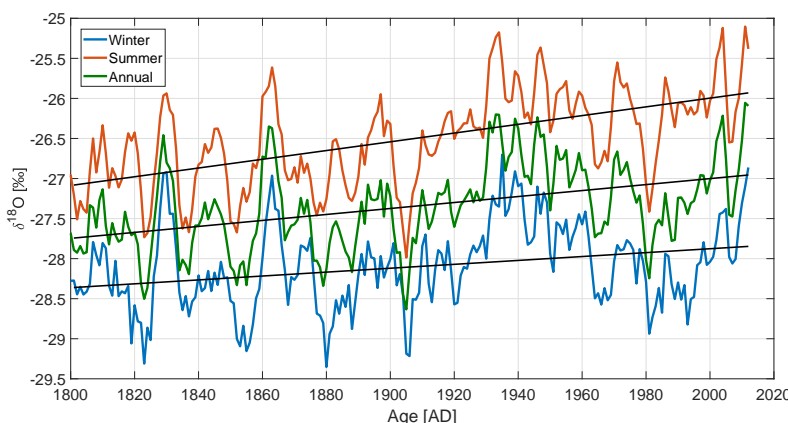

**Figure 3.** Summer (red), winter (blue) and annually averaged (green) $\delta^{18}O$ stacks together with their corresponding linear trends (black lines) for the period 1801–2014. A moving average of 5 years has been applied on all the time series. For the unfiltered series, the reader is referred to Figs. 4, C2 and C3.

# 6   The temperature signature in $\delta^{18}O$

## 6.1   Correlation with instrumental temperature records

The relationship between Renland $\delta^{18}O$ variability and temperature is first investigated by comparing the stacked $\delta^{18}O$ se-

ries with instrumental temperature records. This study uses the nearest and longest temperatures recordings from Greenland (Tasiilaq and Danmarkshavn) and Iceland (Akureyri and Stykkisholmur) - locations are shown in Fig. 1. The Greenland temperature records are available from the Danish Meteorological Institute (http://www.dmi.dk/laer-om/generelt/dmi-publikationer/teknikse-rapporter/) and the Iceland temperatures are available from the Icelandic Met Office (http://en.vedur.is/climatology/data/#a). For the temperature measurements, the seasons have been defined similar to Vinther et al. (2010) with summer ex-

tending from May-October and winter from November-April. Figure 4 shows the annually averaged $\delta^{18}O$ stack together with the annually averaged temperature measurements (winter and summer averages are shown in Fig. C2 and C3). Visually, the past 100 years of summer, winter and annually averaged $\delta^{18}O$ signals of Renland covary with the regional temperature. How-





ever, the years 1830-1910 show periods with both anticorrelation and correlation. Besides the visual covariation, correlation coefficients between the temperature recordings and the $\delta^{18}$O stacks are calculated and shown in Table 4. The correlations with the winter averaged data are in general the lowest while annual and summer signals have similar high correlations at all the sites. The best correlation with the Renland $\delta^{18}$O signal is found for the annual averages at Tasiilaq ($r = 0.50$). Additionally,

5    applying a 5 year moving mean on the $\delta^{18}$O and temperature series increases all the correlations (i.e. the Tasiilaq correlation coefficient increases to r = 0.72).

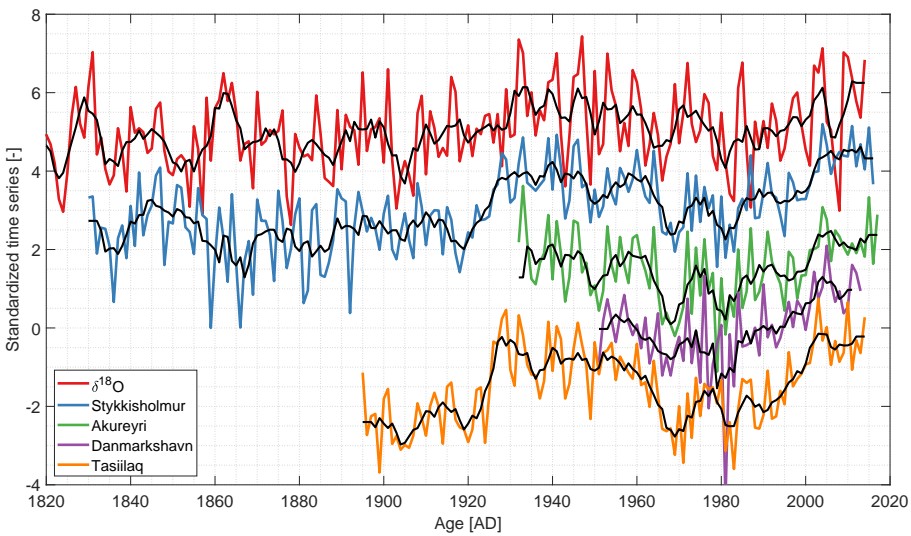

**Figure 4.** Annually averaged $\delta^{18}$O and temperature series. For visualization, the time series have been standardized and shifted vertically. The black curves represent a moving average of 5 years.

**Table 4.** Correlation coefficients between the $\delta^{18}$O stack and instrumental temperature records ($p < 0.05$) on both a 1 year resolution and with a 5 year moving mean applied (in bold).

| Record | Stykkisholmur | Akureyri | Danmarkshavn | Tasiilaq |
|---|---|---|---|---|
| Period | $1830 - 2014$ | $1931 - 2014$ | $1951 - 2014$ | $1895 - 2014$ |
| $r$ winter | $0.29/\mathbf{0.51}$ | $0.30/\mathbf{0.56}$ | $0.21/\mathbf{0.51}$ | $0.41/\mathbf{0.64}$ |
| $r$ summer | $0.40/\mathbf{0.58}$ | $0.45/\mathbf{0.69}$ | $0.30/\mathbf{0.62}$ | $0.37/\mathbf{0.61}$ |
| $r$ annual | $0.48/\mathbf{0.62}$ | $0.40/\mathbf{0.58}$ | $0.35/\mathbf{0.63}$ | $0.50/\mathbf{0.72}$ |

   The high correlation between $\delta^{18}$O and temperature implies that the region's temperature variability is imprinted in the Renland $\delta^{18}$O stack. Conventionally, a simple interpretation in terms of local temperature can then be achieved by using the linear relation between $\delta^{18}$O and temperature. However, this requires that the linear relationship between temperature and

10    $\delta^{18}$O is stable throughout time. In order to examine this, correlation coefficients between Stykkisholmur temperature and



$\delta^{18}$O have been calculated on a 50 year running window and shown in Fig. 5. Here Stykkisholmur is chosen as it has the longest temperature record while we selected a window size of 50 years in order to include enough independent data as the time series have been smoothed with a 5 year moving mean. This analysis indicates that the Stykkisholmur temperature and the $\delta^{18}$O stack only correlates in the period 1910-2014. For winter, summer and annual averages, the average correlation in

the period 1910-2014 is 0.56, 0.65 and 0.66 while it severely reduces to $-0.02$, $-0.02$ and 0.004 in the 1830-1909 period. Thus, the high correlation coefficients presented in Table 4 is only a result of the high correlations in the 1910-2014 period. This could explain why the highest $\delta^{18}$O-temperature correlation was found at Tasiilaq as it only extended back to 1895. Consequently, the regional $\delta^{18}$O-temperature relationship between Renland isotopes and the Iceland temperature record is not constant through time. While it remains unknown if the temperature on Iceland and Renland was similar between 1830-1909,

it is certain that the Renland $\delta^{18}$O variability does not represent the temperature variability at Iceland in said period. Thus, even though the $\delta^{18}$O variability probably reflects the local temperature on Renland, the results show that the decorrelation scale of this $\delta^{18}$O-temperature relationship was different in the 1830-1909 period.

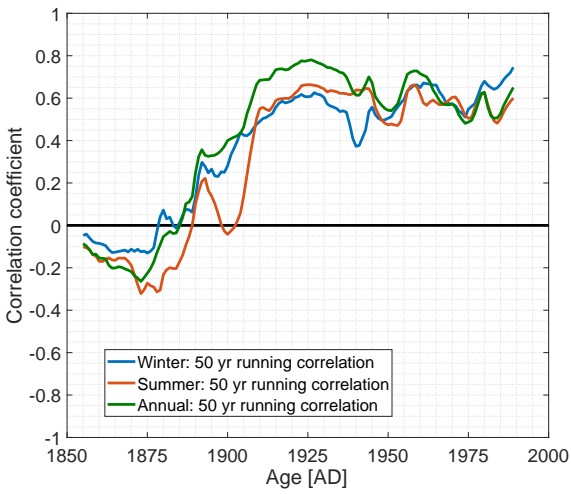

**Figure 5.** Running correlation of 50 years between the Stykkisholmur temperature and the $\delta^{18}$O stack for respectively winter (blue), summer (red) and annual averages (green). Both the $\delta^{18}$O and temperature data were first smoothed with a 5 year moving mean. Each year represents the midpoint of the running window.

### 6.2   Correlation with the HIRHAM5 $2\,m$ temperature output

The spatial extent of the correlation between the $\delta^{18}$O signal and temperature is further investigated by using $2\,m$ temperature

output from the regional climate model, HIRHAM5 (Christensen et al., 2007). This particular HIRHAM5 simulation (Langen et al., 2017) covers the entire Greenlandic region including Iceland. At the lateral boundaries and over the ocean, the model is driven with the European Re-Analysis dataset, ERA-Interim (Dee et al., 2011). This study uses monthly averaged data (1980-2014) on a horizontal resolution of $0.05^{\circ}$ ($\sim 5.5\,km$) converted to summer and winter temperatures by averaging May-October



and November-April, respectively. The RECAP core is used instead of the stacked record as the analysis is on data from the satellite era, which is minimally available in the 1988 cores. The correlation maps are shown in Fig. 6. The results show significant positive correlations between the winter signals of HIRHAM5 $2\,\mathrm{m}$ temperature and the RECAP $\delta^{18}O$. Moreover, the high correlations ($r > 0.5$) that extend over most of Greenland, irrespectively of the ice divide, indicate that the winter

temperature variability over Greenland is imprinted in the Renland $\delta^{18}O$ signal. Results furthermore show that there is no statistically significant correlation between $\delta^{18}O$ and temperature east of Renland in areas regularly covered by sea ice. For the summer and annually averaged signals, the correlations are lower ($r \sim 0.4 - 0.5$) and they only cover the east coast region. This local spatial pattern is consistent with Vinther et al. (2010) who found that the summer averaged $\delta^{18}O$ data from different Greenlandic ice cores were less internally coherent than the corresponding winter data. This could explain why the summer

$\delta^{18}O$ variability of the RECAP core only correlates with the local temperatures on the coast of East Greenland. Moreover, the variance in summer averaged temperatures over Greenland is very low as shown in Fig. 7. The low variance is due to the HIRHAM5 summer temperatures reaching a maxima just below $0^\circ C$ at places with constant ice cover. For instance, Fig. 8 shows the monthly averaged HIRHAM5 temperature from a grid point on Renland where it is evident that the monthly averaged temperature fluctuations during summer are very small. Thus, the small temperature fluctuations can limit the possibility of

interpreting the spatial extent of summer and annual temperature variability imprinted in the $\delta^{18}O$ signal.

All in all, these results support the correlations from Sect. 6.1 that showed high correlations between $\delta^{18}O$ and temperature in the 1910-2014 period.

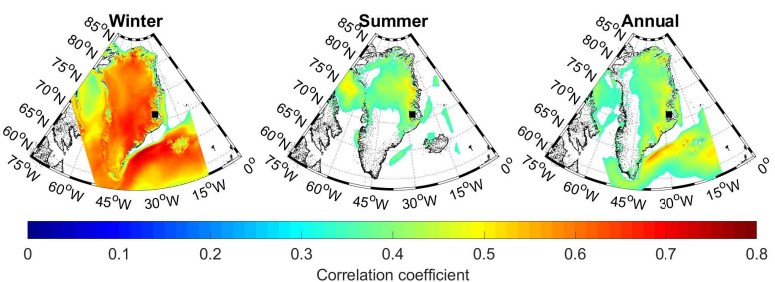

**Figure 6.** Figures showing the correlation between winter (left), summer (middle) and annually (right) averaged RECAP $\delta^{18}O$ and HIRHAM5 temperatures. Only correlations with $p < 0.05$ are shown.

# 7  The North Atlantic Oscillation's imprint on $\delta^{18}O$

The North Atlantic Oscillation (NAO) describes fluctuations in atmospheric pressure at sea level between Iceland and the

Azores. A strengthening and weakening of respectively the low pressure system over Iceland and high pressure system over the Azores control both the direction and strength of westerly winds and storm tracks over the North Atlantic. Such changes





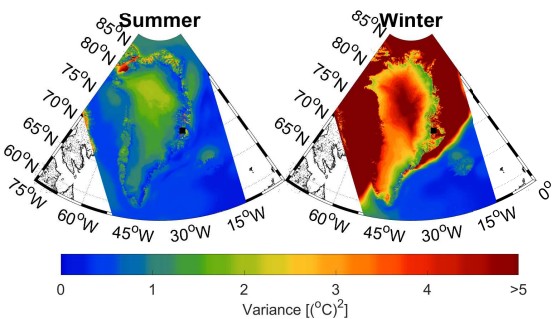

**Figure 7.** Variances of the summer (left) and winter averaged temperatures (right). A maximum variance of $5(^{\circ}\mathrm{C})^2$ is displayed in order to emphasize the small variance in the summer averaged signal.

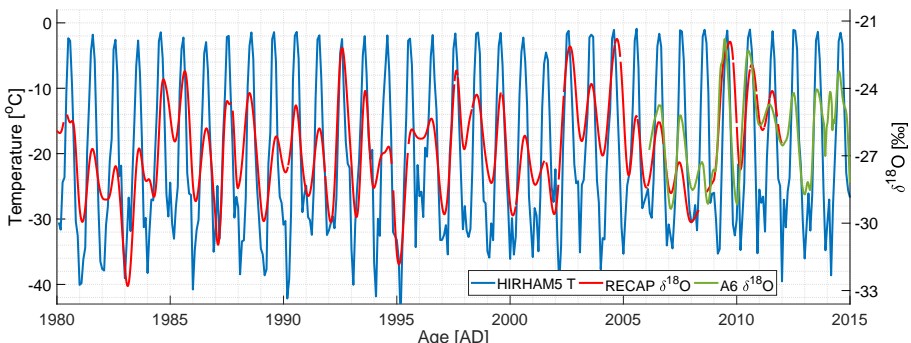

**Figure 8.** Monthly averaged $2\,\mathrm{m}$ temperature from a grid point on Renland (blue curve) plotted together with the forward diffused $\delta^{18}$O from the RECAP ice core (red curve) and A6 snow pit core (green curve).

in the NAO have previously been found to have an imprint on precipitation in western Greenland (Appenzeller et al., 1999). Correspondingly, the winter isotope signal of West and South Greenland ice cores have previously been found to anticorrelate with the atmospheric circulation changes from NAO (Vinther et al., 2003b, 2010). Despite that Vinther et al. (2010) showed that ice cores drilled on the Greenland east coast revealed no connection with the NAO, this study examines said correlation in order to determine if changes in the NAO can be linked to the varying $\delta^{18}$O-temperature relationship.

While the NAO is best described through a principal component analysis of multiple sea level pressure records in the North Atlantic region, this study uses an approximation where the NAO index is based on pressure observations only near the two centers of action of the surface pressure field (the Azores/Iberian Peninsula and Iceland). Such an approximation was carried out by Jones et al. (1997) who reconstructed the NAO variation back to 1821 (and since extended up to present time). This



study uses a slightly modified version of this NAO index by Vinther et al. (2003c) who improved the NAO record in the period 1821-1856 by using extra pressure series.

The connection between the NAO index and seasonally (and annually) averaged $\delta^{18}$O stacks is examined by estimating their correlation. Correlation coefficients have been calculated on 5 year moving averages of the NAO and $\delta^{18}$O stacks and shown in Table 5 (the annual NAO record is plotted in Fig. 9). The level of significance is estimated based on a Monte Carlo routine described in Appendix B. In the complete 1821-2014 period, the summer, winter and annually averaged NAO and $\delta^{18}$O data are uncorrelated with coefficients of $0.01$, $-0.05$ and $0.02$, respectively,. If we instead examine the time before and after the $\delta^{18}$O-temperature correlation terminated (the year 1909), the summer and annually averaged data yield positive correlations of $0.29$ and $0.30$ between 1821-1909 while the winter and annually averaged data yield negative correlations $-0.25$ and $-0.22$ between 1910-2014. Thus, there is a varying relation between the NAO and the $\delta^{18}$O data and the weak $\delta^{18}$O-NAO anticorrelation coincides with a covarying $\delta^{18}$O-temperature relation. However, the weak correlations during 1821-1919 imply that the NAO only can account for around $8-9\%$ of the corresponding $\delta^{18}$O variability. It therefore seems unlikely that respectively strengthening and weakening of the NAO cause changes in the $\delta^{18}$O-temperature relation.

**Table 5.** Correlation coefficients between the $\delta^{18}$O stack and NAO index. Both time series have been smoothed with a 5 year moving mean. Only the numbers in bold are statistically significant ($p < 0.05$).

| Period | $1821 - 1909$ | $1910 - 2014$ | $1821 - 2014$ |
|---|---|---|---|
| $r$ winter | $0.15$ ($p = 0.16$) | **-0.25** ($p = 0.01$) | $-0.05$ ($p = 0.52$) |
| $r$ summer | **0.29** ($p < 0.01$) | $-0.15$ ($p = 0.13$) | $0.01$ ($p = 0.85$) |
| $r$ annual | **0.30** ($p < 0.01$) | **-0.22** ($p = 0.02$) | $0.02$ ($p = 0.82$) |

## 8 The impact of sea ice fluctuations on the stable water isotopes

### 8.1 Fram Strait sea ice export

In this section, it is investigated if there is a connection between the Renland $\delta^{18}$O variability and the sea ice export (SIE) through the Fram Strait (map in Fig. 1). Multi-year sea ice from the Arctic Ocean is exported southward through Fram Strait along the eastern coast of Greenland into the Greenland Sea. Fluctuations in this sea ice volume flux have a direct effect on the amount of open water located east and northeast of Renland. As $\delta^{18}$O is an integrated signal of the hydrological activity along the moisture transport pathway from evaporation source to deposition, the open water which facilitates moist and mild climatic conditions will likely affect the isotopic composition of the precipitation deposited on Renland. Essentially, besides the temperature dependence of isotopic fractionation during local condensation, $\delta^{18}$O contains information about the amount of water mass that is removed from the air during the poleward transport and the continuous contribution of local water mixing with the transported water mass (Noone and Simmonds, 2004).





This analysis uses a Fram Strait SIE record covering the period 1820-2000 reconstructed by Schmith and Hansen (2002). It is an ice volume flux record $\left[\mathrm{km}^3/\mathrm{yr}\right]$ based on historical observations of multi-year sea ice obtained from ship logbooks and ice charts. As the record represents the annual SIE, only the annually averaged $\delta^{18}$O stack is used in the analysis. Figure 9 shows the SIE together with the annually averaged $\delta^{18}$O stack and the RECAP $\mathrm{d}_{\mathrm{xs}}$ record ($\mathrm{d}_{\mathrm{xs}}$ is only available for the RECAP core).

A correlation analysis is carried out in order to quantify any covariation of the records. For a moving average of 5 years applied on the time series, there is an anticorrelation of $-0.54$ ($p < 0.01$) between the annual SIE and $\delta^{18}$O while there is no significant correlation between $\mathrm{d}_{\mathrm{xs}}$ and the SIE ($r = -0.08$). From the correlation analyses, it is clear that $\delta^{18}$O anticorrelates with SIE while it correlates with temperature (Sect. 6.1). In order to examine if these correlations apply simultaneously, correlation coefficients have been calculated on a 50 year running window. The level of significance is estimated based on a Monte Carlo
routine described in Appendix B. The results are plotted in Fig. 10. In the past 100 years, the Stykkisholmur temperature record is found to correlate with Renland $\delta^{18}$O while it (as similar to $\delta^{18}$O) anticorrelates with SIE through Fram Strait. This likely indicates that warm temperatures result in less sea ice that can be exported away from the Arctic Ocean. However, this pattern ceases to exist previous to the early 1900s such that neither the $\delta^{18}$O signal or temperature share any correlation with the SIE. This synchronous decrease in correlation indicates that the uncorrelated $\delta^{18}$O–temperature relation cannot be explained by
dating errors in the ice core chronologies. Furthermore, as discussed in Sect. 7 and shown as running correlations in Fig. 10, the varying $\delta^{18}$O-temperature correlation cannot be a consequence of the NAO controlling the $\delta^{18}$O variability. Moreover, Fig. 10 also shows that changes in local moisture source regions are not traceable through the $\mathrm{d}_{\mathrm{xs}}$-SIE correlation.

In order to examine this $\delta^{18}$O-temperature correlation hiatus, the connection between the SIE anomaly and the $\delta^{18}$O-temperature relation is plotted in Fig. 11 (SIE anomaly is here defined as the deviation from the mean flux). As a 5 year
moving mean has been applied on the time series, only every 5 point is used in the analysis. From the figure, it is clear that on years when the normalized temperature is positive ($T_{\mathrm{norm}} = T - T_{mean}$), there is always a negative SIE anomaly and a high $\delta^{18}$O-temperature correlation of $0.83$. Whereas, for $T_{\mathrm{norm}} < 0$ there is no $\delta^{18}$O-temperature correlation ($r = 0.02$) which coincides with a combination of both positive and negative SIE anomalies. Besides showing that higher temperatures coincide with less multi-year sea ice being transported south (likely due to an already lower extent of sea ice), it appears that lower
temperatures coincide with more fluctuations in the SIE which possibly reduce the $\delta^{18}$O-temperature correlation. These results imply that the $\delta^{18}$O variability can be dominated by other climatic conditions such as SIE, and does not only represent variations in regional temperature for an extended period of time.

## 8.2 Sea ice concentration and sea surface temperature

The Arctic sea ice concentration (SIC) data (fractional ice cover in percentage) from the ERA-Interim reanalysis (Dee et al.,
2011) has been correlated with the RECAP $\delta^{18}$O and $\mathrm{d}_{\mathrm{xs}}$ series and the results are shown in Figs. 12 and 13 (1980-2014). Similar to Sect. 6.2, summer refers to May-October and winter refers to November-April. In the case of $\mathrm{d}_{\mathrm{xs}}$, only the annually averaged data is used as its seasonal components is smeared out after the $\delta^{18}$O and $\delta$D data have been forward diffused. The results show a large patch of anticorrelation between $\delta^{18}$O and SIC in the Baffin Bay area ($r \approx -0.4$) outside West Greenland for both winter and annually averaged data. Presumably, this indicates that the climate at Renland is similar to the





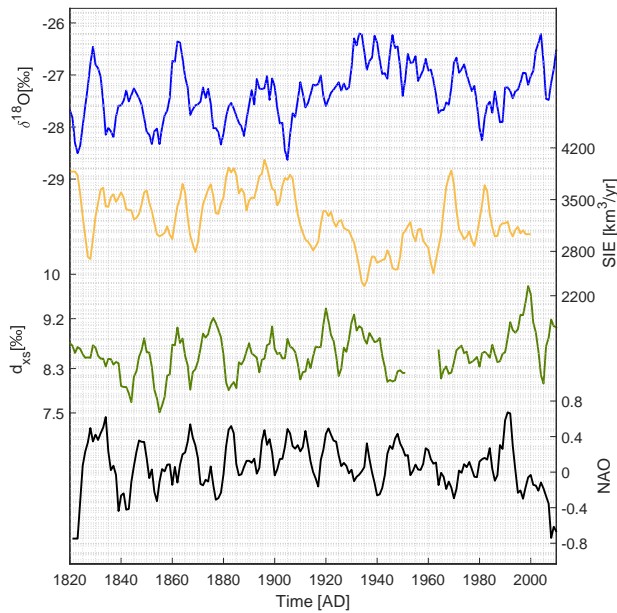

**Figure 9.** Annually averaged $\delta^{18}$O stack (blue curve), Fram Strait SIE (yellow curve), $d_{xs}$ (green curve) and NAO index (black curve). A 5 year moving average has been applied on all the data.

climate at Baffin Bay which controls the advance and retreat of the sea ice extent. A similar connection was found in Sect. 6.2 which showed that winter averaged $\delta^{18}$O signal correlated with temperatures all over Greenland. Resembling anticorrelations between NEEM $\delta^{18}$O and Baffin Bay SIC have previously been found (Steen-Larsen et al., 2011; Zheng et al., 2018). Moreover, the results are consistent with Faber et al. (2017) who found that changes in the Baffin Bay sea ice coverage can impact the

$\delta^{18}$O precipitation over Greenland (by using an atmospheric general circulation model coupled with water isotopologue tracing (isoCAM3)). Furthermore, the analysis shows only a small patch of correlation between the $\delta^{18}$O series and the SIC south of Fram Strait. However, this is not necessarily inconsistent with the significant anticorrelation presented in Sect. 8.1. Possibly, this nuance can be explained by the SIE representing the annual discharge of multi-year sea ice (ice volume flux) while the SIC represents the fractional ice cover in percentage (area).

The connection between the Renland stable water isotopes and the local climate conditions is further investigated by correlating the RECAP $d_{xs}$ signal with the Arctic SIC and sea surface temperature (SST). Figure 13 shows that there exists small patches of positive correlation patterns between the $d_{xs}$ signal and the SIC in the Arctic Ocean and south of Baffin Bay. As these areas are very small, it is difficult to evaluate the connection between the extent of SIC and $d_{xs}$ at Renland. The $d_{xs}$ signal is further examined by checking if it reflects the local SST variability. This has been done by correlating the $d_{xs}$ signal with the

SST data in the Arctic region from ERA-Interim data (1980-2014). From Fig. 13, it is evident that there barely exists patches



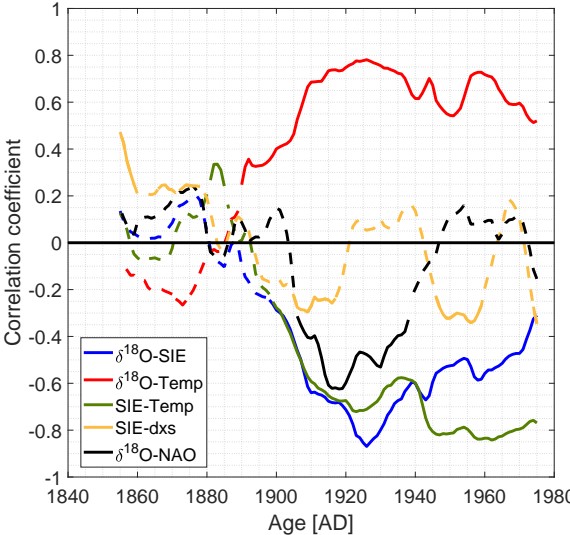

**Figure 10.** Running correlations of 50 years between Stykkisholmur temperature and the $\delta^{18}O$ stack (red), SIE and the $\delta^{18}O$ stack (blue), SIE and Stykkisholmur temperature (green), the $\delta^{18}O$ stack and the NAO index (black) and $d_{xs}$ with SIE (yellow). The solid lines represent significant correlation ($p < 0.05$) while the dashed lines are insignificant $p > 0.05$. Each year represents the midpoint of the running window.

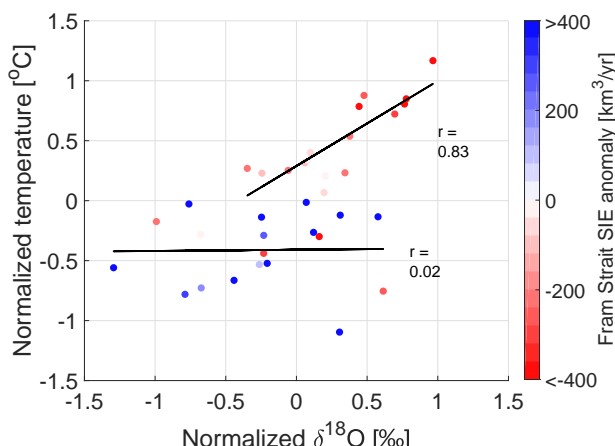

**Figure 11.** Normalized annual temperature plotted with respect to normalized annual $\delta^{18}O$ where colors indicate strength of the Fram Strait sea ice export anomaly. A 5 year moving average has been applied to all the time series but only every 5 point is displayed and used in the analysis. The solid black lines represent linear fits between $\delta^{18}O$ and temperature for positive and negative temperature anomalies.

with significant correlation. Thus, it is difficult to assess whether the RECAP $d_{xs}$ record directly reflects the local SST or SIC variability during the 1980-2014 period. More analysis on what controls the Renland $d_{xs}$ signal is needed in future research.



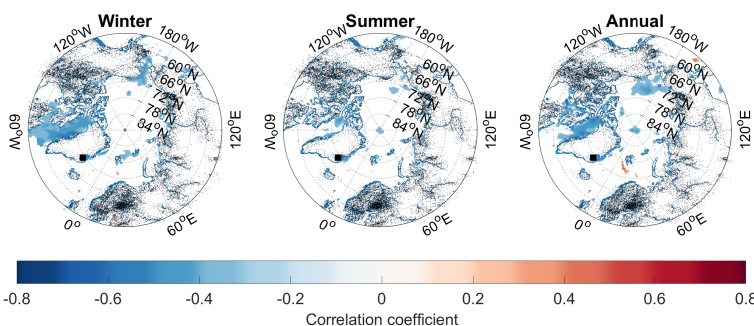

**Figure 12.** Maps showing the correlation coefficients between the ERA-Interim sea ice concentration and the RECAP $\delta^{18}O$ data for the 1980-2014 period ($p < 0.05$).

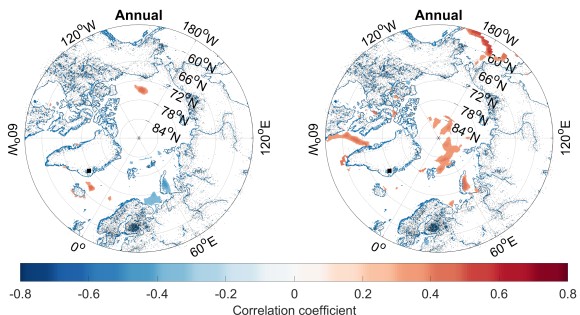

**Figure 13.** Maps showing the $d_{xs}$-SST (left) and $d_{xs}$-SIC correlation coefficients between annually averaged data from RECAP and ERA-Interim covering the 1980-2014 period ($p < 0.05$).

## 9 Discussion

The analysis showed that the strength of the SIE have varied in the past and that its fluctuations could be connected with the regional $\delta^{18}O$-temperature relationship. Despite the apparent connection, this study has not proved any causality between the $\delta^{18}O$-temperature relation and the Fram Strait SIE. Still, a proposed hypothesis for this connection is that the fluctuating SIE conditions during cold years impose changes in the location of the local moisture sources which suppress the imprint of Iceland temperature variability in Renland $\delta^{18}O$. It is likely that this connection has its strongest impact on ice cores drilled at the coastal regions near sea ice as Vinther et al. (2010) found that the $\delta^{18}O$ records of Greenlandic ice cores drilled in South and Central Greenland correlated well with a Southwest Greenland instrumental temperature series in the period 1785-



1980 (Vinther et al., 2006). With reference to this temperature series, Fig. C4 shows that these temperatures do not have a stable linear covariation with the Renland $\delta^{18}$O stack (winter averages are here chosen as that constitutes the longest and most homogeneous record). Besides that Renland obviously is located far away from the Southwest Greenland instrumental temperature stations, this contrariety might result from the isotope distillation process being more manifested as a temperature

variability in the $\delta^{18}$O signal when the precipitation has journeyed further and risen in altitude more than that of the coastal region, further depleting the $\delta^{18}$O signal. In order to evaluate this hypothesis, more studies using isotope-enabled modeling are needed. The impact of changes in sea ice on the Arctic $\delta^{18}$O precipitation has previously been investigated by Faber et al. (2017) who found that the $\delta^{18}$O precipitation on Greenland only responded to perturbations of the Baffin Bay sea ice coverage. However, they used a horizontal resolution of $\sim 1.4^{\circ} \times 1.4^{\circ}$ which barely resolved the Renland Ice Cap of $\sim 1200\,\mathrm{km}^2$. Thus, a

further examination of how changes in sea ice extent is connected with the coastal Greenlandic precipitation on a higher spatial resolution grid is essential in order to evaluate this hypothesis.

Moreover, while this study found that the Renland $\delta^{18}$O signal anticorrelated with variations in the sea ice extent outside West Greenland (Sect. 8.2), a similar pattern was found with the HIRHAM5 temperature correlations presented in Sect. 6.2. It is therefore likely that the connection represents a reduced sea ice coverage due to increasing temperatures rather than an

actual interconnection between Renland $\delta^{18}$O and Baffin Bay sea ice.

## 10   Conclusions

This study found that by quantifying the mean signal to noise variance ratios, a robust seasonal $\delta^{18}$O signal (1801-2014) could be extracted by stacking three ice cores from Renland. This $\delta^{18}$O stack was correlated with instrumental temperature records from East Greenland and Iceland and with the HIRHAM5 $2\,\mathrm{m}$ temperature output. Results showed that there were

high correlations between $\delta^{18}$O and regional temperatures on both a seasonal and annual scale between 1910-2014. A similar anticorrelation was found between the $\delta^{18}$O stack and the amount of sea ice exported through Fram Strait. However, both correlations stopped in the 1830-1909 period. The results indicated that the varying regional temperature variability in the $\delta^{18}$O signal could not be explained by the North Atlantic Oscillation. Instead, the linear $\delta^{18}$O-temperature relation depended on whether the temperatures were warmer or colder than the temperature anomaly. Warm years were associated with a high

correlation and accompanied by less sea ice transported south along the coast while cold years were associated with zero correlation that accompanied a fluctuating amount of sea ice along the coast. These results implied that changes in the extent of open water outside Renland might affect the local moisture conditions. Hence, greater sea ice flux along the coast of Greenland may suppress the Iceland temperature signature in the d18O signal; however, this was not confirmed by correlations between $\mathrm{d_{xs}}$ and sea surface temperature in the Arctic region. Thus, more high resolution isotope-enabled modeling focused on the

effect of Arctic sea ice on coastal precipitation are needed in order to quantify this process.

These results have implications for ice core temperature reconstructions based on the linear relationship between $\delta^{18}$O variability and local temperature records. For Renland, the linear $\delta^{18}$O-temperature relationship was unstable with time which implied that the annual-to-decadal variability of $\delta^{18}$O measured in an ice core could not be directly attributed to temperature



variability. Similar conditions might apply for other ice cores drilled in the vicinity of a fluctuating sea ice cover. This reinforces the interpretation that $\delta^{18}$O is an integrated signal of all the hydrological activity that a vapor mass experiences from the evaporation at the source to its condensation at the drill site.

**Appendix A: Firn diffusivity**

This study uses the firn diffusivity parameterization of Johnsen et al. (2000):

$$D(\rho) = \frac{m\,p\,D_{\mathrm{a}i}}{R\,T\,\alpha_{\mathrm{i}}\,\tau}\left(\frac{1}{\rho} - \frac{1}{\rho_{\mathrm{ice}}}\right) \tag{A1}$$

which depends on the molar weight of water ($m$), the saturation vapor pressure ($p$), diffusivity of water vapor ($D_{\mathrm{a}i}$), the molar gas constant ($R$), the site temperature ($T$), the ice–vapor fractionation factor ($\alpha_i$) and the firn air tortuosity ($\tau$). Similar to Johnsen et al. (2000) and subsequently used in Simonsen et al. (2011); Gkinis et al. (2014); Holme et al. (2018), we used the

following definitions which can be parameterized through annual mean surface temperature, annual accumulation rate, surface pressure and density ($\rho$):

    – Saturation vapor pressure over ice (Pa) (Murphy and Koop, 2006):

$$p = \exp\left(9.5504 - \frac{5723.265}{T} + 3.530\ln(T) - 0.0073\,T\right) \tag{A2}$$

– $D_{\mathrm{a}i}$: diffusivity of water vapor (for isotopologue $i$) in air (m$^2$s$^{-1}$). For the diffusivity of the abundant isotopologue water vapor $D_{\mathrm{a}}$ (Hall and Pruppacher, 1976):

$$D_{\mathrm{a}} = 2.1\cdot 10^{-5}\left(\frac{T}{T_o}\right)^{1.94}\left(\frac{P_o}{P}\right) \tag{A3}$$

    with $P_{\mathrm{o}} = 1$ Atm, $T_{\mathrm{o}} = 273.15$ K and $P, T$ the ambient pressure (Atm) and temperature (K). Additionally from Merlivat and Jouzel (1979) $D_{a^2\mathrm{H}} = 0.9755 D_{\mathrm{a}}$ and $D_{a^{18}\mathrm{O}} = 0.9723 D_{\mathrm{a}}$

– $R$: molar gas constant $R = 8.3144\,\mathrm{m^3 Pa K^{-1} mol^{-1}}$

    – $\alpha_i$: Ice – Vapor fractionation factor. we use the formulations by Majoube (1970) and Merlivat and Nief (1967) for $\alpha_{\mathrm{s/v}}^{\delta\mathrm{D}}$ and $\alpha_{\mathrm{s/v}}^{\delta^{18}\mathrm{O}}$ respectively.

$$\alpha_{\mathrm{Ice/Vapor}}\left(^2\mathrm{H}/^1\mathrm{H}\right) = 0.9098\exp(16288/T^2) \tag{A4}$$

$$\alpha_{\mathrm{Ice/Vapor}}\left(^{18}\mathrm{O}/^{16}\mathrm{O}\right) = 0.9722\exp(11.839/T) \tag{A5}$$

$\tag{A6}$



- $\tau$: The firn tortuosity (Schwander et al., 1988):

$$\frac{1}{\tau} = 1 - b \cdot \left(\frac{\rho}{\rho_{\text{ice}}}\right)^2 \quad \rho \leq \frac{\rho_{\text{ice}}}{\sqrt{b}}, \ b = 1.3 \tag{A7}$$

Based on Eq. (A7), $\tau \to \infty$ for $\rho > 804.3\,\mathrm{kg\,m^{-3}}$

## Appendix B: Significance analysis

In this study, time series have often been smoothed with a 5 year moving mean before estimating their correlation. Potentially, this results in artificially improved correlation coefficients as a moving mean is a low-pass filter. It is therefore necessary to quantify the significance of the linear relationship ($p-$value) by running a Monte Carlo simulation. This study test said significance by examining what correlation coefficients we would estimate if we had random generated data instead of the $\delta^{18}$O signal (following the procedure proposed by Macias-Fauria et al. (2011)). For simplicity, this section refers to the correlation between

$\delta^{18}$O and temperature while it applies equally for all types of time series.

Synthetic data are created by generating time series with the same power spectrum as the $\delta^{18}$O signal. This study uses a method outlined in Ebisuzaki (1977) that is based on a random resampling of the $\delta^{18}$O signal in the frequency domain. The synthetic time series is then found by taking the inverse fast Fourier transform of the shuffled signal. This procedure retains the same autocorrelation as the input time series hereby mimicking the influence of a 5 year moving mean applied on data.

This procedure is simulated 1000 times. For each iteration, the correlation coefficient between the synthetic $\delta^{18}$O series and the temperature series is calculated. From this Monte Carlo routine, an empirical probability distribution function that describes the relation between synthetic generated data and the temperature series is obtained. From this distribution, it is possible to compute the $p-$value which describes how probable it is that the correlation between $\delta^{18}$O and temperature is significantly different from that of the synthetic $\delta^{18}$O and temperature. In this study, $p-$values below $0.05$ are considered

statistically significant.

## Appendix C: Figures

*Competing interests.*  The authors declare that they have no conflict of interest.

*Acknowledgements.*  The RECAP ice coring effort was financed by the Danish Research Council through a Sapere Aude grant, the NSF through the Division of Polar Programs, the Alfred Wegener Institute, and the European Research Council under the European Community's
Seventh Framework Programme (FP7/2007-2013)/ ERC grant agreement 610055 through the Ice2Ice project and the Early Human Impact project (267696). The authors acknowledge the support of the Danish National Research Foundation through the Centre for Ice and Climate at the Niels Bohr Institute (Copenhagen, Denmark).





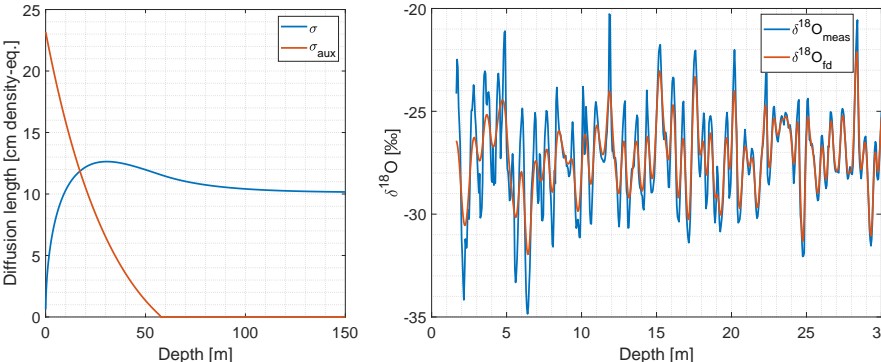

**Figure C1.** Left: Modeled firn diffusion with depth ($\sigma$; blue) and calculated auxiliary diffusion length that should be applied on the measured $\delta^{18}$O data ($\sigma_{aux}$; red). After the pore close–off ($\rho_{pc} = 804.3 \, \mathrm{kg \, m^{-3}}$), $\sigma_{aux} = 0$ as $\sigma$ just changes due to the compaction of firn. Right: The measured $\delta^{18}$O data (blue) and the forward-diffused $\delta^{18}$O data (red) for the 1988 M core.

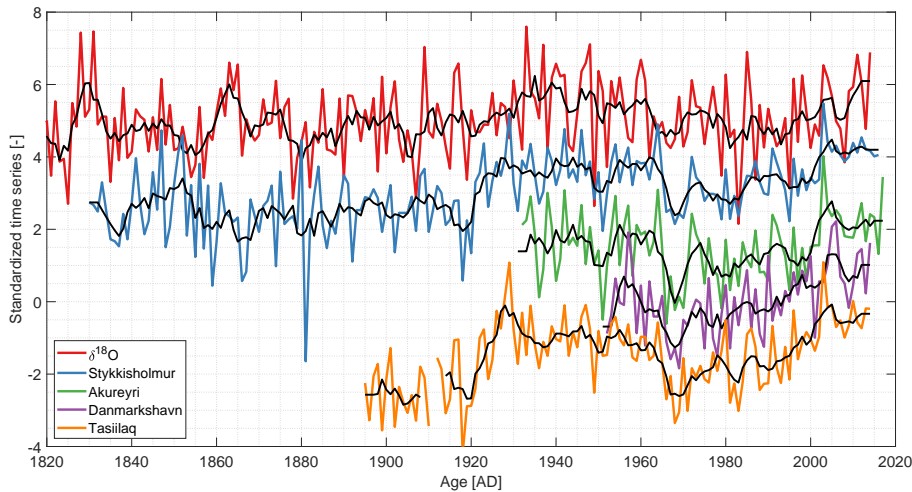

**Figure C2.** Winter averaged $\delta^{18}$O and temperature series. For visualization, the time series have been standardized and shifted vertically. The black curves represent a moving average of 5 years.

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





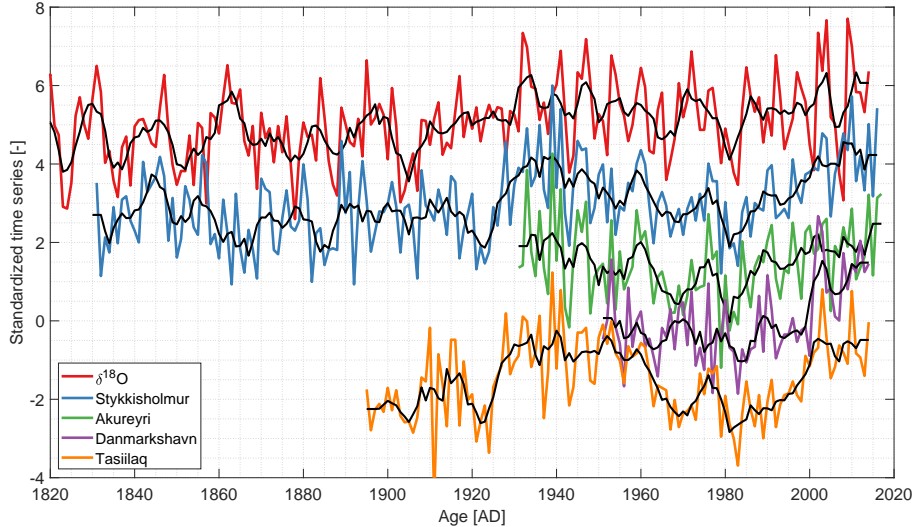

**Figure C3.** Summer averaged $\delta^{18}$O and temperature series. For visualization, the time series have been standardized and shifted vertically. The black curves represent a moving average of 5 years.

Christensen, O. B., Drews, M., Christensen, J. H., Dethloff, K., Ketelsen, K., Hebestadt, I., and Rinke, A.: The HIRHAM Regional Climate Model Version 5, Technical report 06-17, Danish Climate Centre, DMI, 2007.

Cuffey, K. M. and Steig, E. J.: Isotopic diffusion in polar firn: implications for interpretation of seasonal climate parameters in ice-core records, with emphasis on central Greenland, Journal of Glaciology, 44, 273–284, 1998.

Dansgaard, W.: The $^{18}$O-abundance in fresh water, Geochimica et Cosmochimica Acta, 6, 241–260, 1954.

Dansgaard, W.: Stable isotopes in precipitation, Tellus B, 16, 436–468, 1964.

Dee, D. P., Uppala, S. M., Simmons, A., Berrisford, P., Poli, P., Kobayashi, S., Andrae, U., Balmaseda, M., Balsamo, G., Bauer, d. P., et al.: The ERA-Interim reanalysis: Configuration and performance of the data assimilation system, Quarterly Journal of the royal meteorological society, 137, 553–597, 2011.

Ebisuzaki, W.: A method to estimate the statistical significance of a correlation when the data are serially correlated, Journal of Climate, 10, 2147–2153, 1977.

Ekaykin, A. A., Vladimirova, D. O., Lipenkov, V. Y., and Masson-Delmotte, V.: Climatic variability in Princess Elizabeth Land (East Antarctica) over the last 350 years, Climate of the Past, 13, 61–71, 2017.

Epstein, S., Buchsbaum, R., Lowenstam, H., and Urey, H.: Carbonate-water isotopic temperature scale, Geological Society of America

Bulletin, 62, 417, 1951.

Faber, A.-K., Vinther, B. M., Sjolte, J., and Pedersen, R.: How does sea ice influence $\delta^{18}$O of Arctic precipitation?, Atmospheric Chemistry and Physics, 17, 5865–5876, 2017.

Gkinis, V., Simonsen, S. B., Buchardt, S. L., White, J. W. C., and Vinther, B. M.: Water isotope diffusion rates from the NorthGRIP ice core for the last 16,000 years - glaciological and paleoclimatic implications, Earth and Planetary Science Letters, 405, 2014.





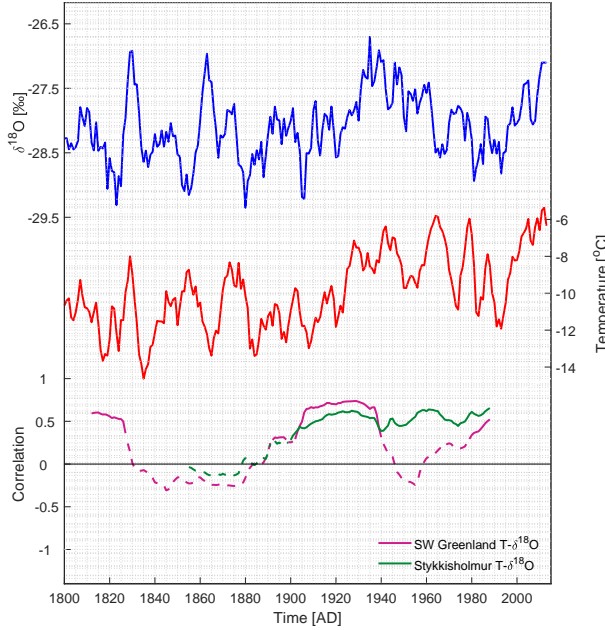

**Figure C4.** Top: 5 year moving average of the winter averaged $\delta^{18}$O stack. Mid: 5 year moving average of the December-January-February averaged Southwest Greenland temperatures from Vinther et al. (2006). Bottom: 50 year running correlations between $\delta^{18}$O and Southwest Greenland (magenta) and $\delta^{18}$O and Stykkisholmur (green). Each year represents the midpoint of the running window. Solid lines are significant correlations and dashed lines are insignificant ($p > 0.05$).

Hall, W. D. and Pruppacher, H. R.: The Survival of Ice Particles Falling from Cirrus Clouds in Subsaturated Air, Journal of the Atmospheric Sciences, 33, 1995–2006, 1976.

Herron, M. M. and Langway, C. C.: Firn Densification: An Empirical Model, Journal of Glaciology, 25, 1980.

Holme, C., Gkinis, V., and Vinther, B. M.: Molecular diffusion of stable water isotopes in polar firn as a proxy for past temperatures, Geochemica et Cosmochimica Acta, 225, 128–145, 2018.

Johnsen, S., Clausen, H. B., Cuffey, K. M., Hoffmann, G., Schwander, J., and Creyts, T.: Diffusion of stable isotopes in polar firn and ice: the isotope effect in firn diffusion, Physics of Ice Core Records, pp. 121–140, 2000.

Johnsen, S. J.: Stable Isotope Homogenization of Polar Firn and Ice, Isotopes and Impurities in Snow and Ice, pp. 210–219, 1977.

Johnsen, S. J., Dansgaard, W., and White, J.: The origin of Arctic precipitation under present and glacial conditions, Tellus, 41B, 452–468, 1989.

Johnsen, S. J., Clausen, H. B., Dansgaard, W., Gundestrup, N. S., Hansson, M., Jonsson, P., Steffensen, J. P., and Sveinbjornsdottir, A. E.: A "deep" ice core from East Greenland, Meddelelser om Groenland, Geoscience, 29, 3–22, 1992.

Johnsen, S. J., Clausen, H. B., Dansgaard, W., Gundestrup, N. S., Hammer, C. U., Andersen, U., Andersen, K. K., Hvidberg, C. S., Dahl-Jensen, D., Steffensen, J. P., Shoji, H., Sveinbjornsdottir, A. E., White, J., Jouzel, J., and Fisher, D.: The $\delta^{18}$O record along the Greenland



Ice Core Project deep ice core and the problem of possible Eemian climatic instability, Journal of Geophysical Research, 102, 26 397—26 410, https://doi.org/10.1029/97JC00167, 1997.

Johnsen, S. J., Dahl-Jensen, D., Gundestrup, N., Steffensen, J. P., Clausen, H. B., Miller, H., Masson-Delmotte, V., Sveinbjornsdottir, A. E., and White, J.: Oxygen isotope and palaeotemperature records from six Greenland ice-core stations: Camp Century, Dye-3, GRIP, GISP2, Renland and NorthGRIP, Journal of Quaternary Science, 16, 2001.

Jones, P. D., Jonsson, T., and Wheeler, D.: Extension to the North Atlantic oscillation using early instrumental pressure observations from Gibraltar and south-west Iceland, International Journal of Climatology, 17, 1433–1450, 1997.

Jones, P. D., Osborn, T. J., Briffa, K. R., Folland, C. K., Horton, E. B., Alexander, L. V., Parker, D. E., and Rayner, N. A.: Adjusting for sampling density in grid box land and ocean surface temperature time series, Journal of Geophysical Research, 106, 3371–3380, 2001.

Jouzel, J. and Merlivat, L.: Deuterium and oxygen 18 in precipitation: modeling of the isotopic effects during snow formation, Journal of Geophysical Research-Atmospheres, 89, 11 749 – 11 759, 1984.

Jouzel, J., Alley, R. B., Cuffey, K. M., Dansgaard, W., Grootes, P., Hoffmann, G., Johnsen, S. J., Koster, R. D., Peel, D., Shuman, C. A., Stievenard, M., Stuiver, M., and White, J.: Validity of the temperature reconstruction from water isotopes in ice cores, Journal Of Geophysical Research-Oceans, 102, 26 471–26 487, 1997.

Klein, E. S. and Welker, J. M.: Influence of sea ice on ocean water vapor isotopes and Greenland ice core records, Geophysical Research Letters, 43, 12,475–b12,483, 2016.

Langen, P. L., Fausto, R. S., Vandecrux, B., Mottram, R. H., and Box, J. E.: Liquid Water Flow and Retention on the Greenland Ice Sheet in the Regional Climate Model HIRHAM5: Local and Large-Scale Impacts, Front. Earth Sci., 2017.

Macias-Fauria, M., Grinsted, A., Helama, S., and Holopainen, J.: Persistence matters: Estimation of the statistical significance of paleoclimatic reconstruction statistics from autocorrelated time series, Dendrochronologia, 30, 179–187, 2011.

Majoube, M.: Fractionation factor of $^{18}$O between water vapour and ice, Nature, 226, 1970.

Merlivat, L. and Jouzel, J.: Global Climatic Interpretation of the Deuterium-Oxygen 18 Relationship for Precipitation, Journal of Geophysical Research, 84, 1979.

Merlivat, L. and Nief, G.: Fractionnement Isotopique Lors Des Changements Detat Solide-Vapeur Et Liquide-Vapeur De Leau A Des Temperatures Inferieures A 0 Degrees C, Tellus, 19, 122–127, 1967.

Mook, J.: Environmental Isotopes in the Hydrological Cycle Principles and Applications, International Atomic Energy Agency, 2000.

Murphy, D. M. and Koop, T.: Review of the vapour pressures of ice and supercooled water for atmospheric applications, Q.J.R. Meteorol. Soc., 131, 1539–1565, 2006.

Noone, D. and Simmonds, I.: Sea ice control of water isotope transport to Antarctica and implications for ice core interpretation, Journal of Geophysical Research, 109, 2004.

Osborn, T., Briffa, K. R., and Jones, P. D.: Adjusting Variance for sample-size in tree-ring chronologies and other regional mean time series, Dendrochronologies, 15, 1997.

Schmith, T. and Hansen, C.: Fram Strait Ice Export during the Nineteenth and Twentieth Centuries Reconstructed from a Multiyear Sea Ice Index from Southwestern Greenland, Journal of Climate, 16, 2782–2791, 2002.

Schwander, J., Stauffer, B., and Sigg, A.: Air mixing in firn and the age of the air at pore close-off, Annals of Glaciology, pp. 141 –145, 1988.

Simonsen, M. F., Baccolo, G., Borunda, A., Delmonte, B., Goldstein, S., Grindsted, A., Kjaer, H. A., Sowers, T., Svensson, A., Vinther, B. M., Winckler, G., Winstrup, M., and Vallelonga, P.: Ice core dust particle size reveals past glacier extent in East Greenland, In review in Nature Communication, ., 2018, in review.





Simonsen, S. B., Johnsen, S. J., Popp, T. J., Vinther, B. M., Gkinis, V., and Steen-Larsen, H. C.: Past surface temperatures at the NorthGRIP drill site from the difference in firn diffusion of water isotopes, Climate of the Past, 7, 2011.

Steen-Larsen, H. C., Masson-Delmotte, V., Sjolte, J., Johnsen, S. J., Vinther, B. M., Bréon, F., Clausen, H. B., Dahl-Jensen, D., Falourd, S., Fettweis, X., Gallée, H., Jouzel, J., Kageyama, M., Lerche, H., Minster, B., Picard, G., Punge, H. J., Risi, C., Salas, D., Schwander, J.,

Steffen, K., Sveinbjornsdottir, A. E., Svensson, A., and White, J.: Understanding the climatic signal in the water stable isotope records from the NEEM shallow firn/ice cores in northwest Greenland, Journal of Geophysical Research, 116, 2011.

Steig, E. J., Ding, Q., White, J. W. C., Kuttel, M., Rupper, S. B., Neumann, T. A., Neff, P. D., Gallant, A. J. E., Mayewski, P. A., Taylor, K. C., Hoffmann, G., Dixon, D. A., Schoenemann, S., M., M. B., Schneider, D. P., Fudge, T. J., Schauer, A. J., Teel, R. P., Vaughn, B., Burgener, L., Williams, J., and Korotkikh, E.: Recent climate and ice-sheet change in West Antarctica compared to the past 2000 years,

Nature Geoscience, 6, 2013.

Vinther, B. M.: Seasonal $\delta^{18}$O Signals in Greenland Ice Cores, Master's thesis, University of Copenhagen, Denmark, 2003a.

Vinther, B. M., Johnsen, S. J., Andersen, K. K., Clausen, H. B., and Hansen, A. W.: NAO signal recorded in the stable isotopes of Greenland ice cores, Geoph. Res. Lett., 30, 2003b.

Vinther, B. M., Andersen, K. K., Hansen, A. W., Schmidth, T., and Jones, P. D.: Improving the Gibraltar/Reykjavik NAO Index, Geoph. Res.

Lett., 30, 2003c.

Vinther, B. M., Andersen, K. K., Jones, P. D., Briffa, K. R., and Cappelen, J.: Extending Greenland temperature records into the late eighteenth century, Journal of Geophysical Research, 111, 2006.

Vinther, B. M., Buchardt, S. L., Clausen, H. B., Dahl-Jensen, D., Johnsen, S. J., Fisher, D. A., Koerner, R. M., Raynaud, D., Lipenkov, V., Andersen, K. K., Blunier, T., Rasmussen, S. O., Steffensen, J. P., and Svensson, A. M.: Holocene thinning of the Greenland ice sheet,

Nature, 461, 2009.

Vinther, B. M., Jones, P. D., Briffa, K. R. and. Clausen, H. B., Andersen, K. K., Dahl-Jensen, D., Johnsen, S. J., and Clausen, H. B.: Climatic signals in multiple highly resolved stable isotope records from Greenland, Quaternary Science Reviews, 29, 522–538, 2010.

Winstrup, M., Svensson, A. M., Rasmussen, S. O., Winther, O., Steig, E. J., and Axelrod, A. E.: An automated approach for annual layer counting in ice cores, Clim. of the Past, 8, 1881–1895, 2012.

Zheng, M., Sjolte, J., Alolphi, F., Vinther, B. M., Steen-Larsen, H. C., Popp, T. J., and Muscheler, R.: Climate information preserved in seasonal water isotope at NEEM: relations with temperature, circulation and sea ice, Clim. of the Past, 14, 1067–1078, 2018.