# Peer review of "Varying regional $\delta^{18}$ O–temperature relationship in high resolution stable water isotopes from East Greenland"

_Climate of the Past, 2018_

## Referee Comment (RC1) · Divine (Referee) · 9 Jan 2019

The comment was uploaded in the form of a supplement:
https://www.clim-past-discuss.net/cp-2018-169/cp-2018-169-RC1-supplement.pdf

---

## Referee Comment (RC2) · Anonymous Referee #2 · 15 Jan 2019

The authors demonstrate new high-resolution stable water isotope records from Renland peninsula (East Greenland) and make an attempt to establish the relationship between the isotopic composition and the regional instrumental air temperature data. They show that the temperature in the condensation site is not the only driver, the other factors like sea ice concentration in the nearby ocean and possibly regional circulation pattern (NAO) being important. Thus the complex nature of the isotopic composition of the atmospheric precipitation is clearly demonstrated. The obtained results are important for the interpretation of the isotpic content of the fossil precipitation retrieved as firn and ice cores by means of deep drilling of the polar ice sheets.

[Figure]

To my opinion, the manuscript can be published with minor revisions, as listed below:

Page 1, lines 6-7: consider rewording, e.g. "and examine which amount of this variability could be attributed..."

Page 2, line 2: from the context of this paragraph it's clear that you are talking about polar ice sheets, so it's better to write "polar ice sheets" instead of "ice caps" here.

Page 3, line 3 - the resolution of the map in Figure 1 does not really allow to see it.

Page 3, line 5 and below - Inland or inland?

Page 3, line 14 - "as we here have" sounds awkward for me.

Page 9, line 2 - in order to filter out

Page 14, line 7 - unwanted comma in the end of the sentence

Page 14, line 11 - should be 1821-1909?

Page 15, line 21, and Figure 11 - the term "normalized" is usually defined by (value - mean)/standard_deviation. What is shown here is a temperature anomaly (= deviation from mean).

---

## Short Comment (SC1) · 18 Jan 2019

Dear Authors,

Congratulations to the interesting study. I've two points for this short comment.

1, Can you make available the raw d18O series of the Renland ice cores? or the annualized d18O records? I cannot find any link either to a data repository or to a supplementary source where the ice core derived d18O data could be available.

2, Changes in the seasonal distribution of precipitation could be also a potential explanation for temporal changes is d18O-temperature relation. Even seasonal distribu-

tion of precipitation since 1910 might explain the good correlation with annual mean air temperature. However, an increased seasonal contrast in the intraanual distribution of precipitation before 1910 might introduce a seasonal bias to the depositional d18O record and it can disturb the correlation with the annual mean air temperature. It can be tested if amount-weigthed mean temperatures could be calculated and involved also into the correlation analysis. A recent study from Polish Polar Research (https://content.sciendo.com/view/journals/popore/38/2/article-p105.xml) can be potentially considered in this idea.

best regards, Zoltan Kern

---

## Author Comment (AC1) · 24 Apr 2019

We thank the reviewers for their comments and suggestions which have helped improving the manuscript. Besides minor corrections to the text, we have, as suggested by reviewer 1, extended the analysis by including sea ice edge anomaly data in the study (Section 8.2 in the revised manuscripts). A brief summary of these results is presented in this document.

Specific comments to each reviewer is provided below. The reviewer's comments are written in bold and our answers are written with regular font.

**Reviewer 1**

**In this manuscript the authors demonstrates a time dependent correlation of a stack of three forward-diffused d18O records from Renland ice cap with a series of longer temperature records from the Greenland region. The authors relate a lack of correlation for the pre- 1910 period with the increased sea ice extent in the Greenland and Iceland seas (more precisely, Fram Strait sea ice export) and sea ice effects on moisture pathways/ distillation histories.**

**In general the paper is clearly written and results are well presented. My only moderate concern is related with the use of index based sea ice export reconstruction as a sole indicator of sea ice anomalies in the Greenland and Iceland seas. From the point of view of direct physical controls on the local d18O in precipitation, these are sea ice extent anomalies along the moisture pathways that are the more obvious candidates rather than sea ice export which integrates extent together with other relevant variables (drift velocity, spatial ice thickness distribution etc). The authors already refer to Noone and Simmonds that discusses the driving mechanisms for sea ice extent - d18O relationship. Therefore in addition to the reconstruction of Schmith&Hansen, the authors may consider using seasonal ice extent anomalies from Divine&Dick (2006), doi:10.1029/2004JC002851, directly, the data are found in https://nsidc.org/data/g02169.**

This is a very good point and we thank the reviewer for making us aware of the sea ice edge anomaly data from Divine and Dick (2006). In the revised manuscript (Section 8.2), we have included the seasonal sea ice edge anomaly data (Greenland Sea) in our analysis with the Renland d18O signal (a summary of our results is provided below).

Notably, our results show that the sea ice edge and d18O data only are weakly correlated (and uncorrelated with Fram Strait sea ice export). While we think part of this can be explained by the two sea ice proxies representing slightly different features (one is a position metric and the other is a volume flux), we did not expect this difference in the results. It is possible that the d18O-sea ice export correlation is strengthened through the sea ice export's dependence on its drifting velocity (winters exert stronger northerly winds (Steffen and Box, 2001)). This feature will increase the sea ice export additionally independent on the actual amount of sea ice. While this could enhance the correlation, it does not explain why both the d18O and the sea ice export become uncorrelated

synchronously with their correlation with the Iceland temperature (Sect. 8.1). This indicates that the d18O-sea ice export connection still is more complex than a wind influence.

Furthermore, despite that d18O and the sea ice anomaly data not are linearly connected, the sea ice edge anomaly data is still convenient as it shows how the sea ice extent in the Greenland Sea was much greater prior to the 1910s. This strengthens our hypothesis stating that changes in the sea ice extent influences the spatial extent of the d18O-temperature relationship.

**In general, the manuscript deserves to be published after these moderate modifications to the content if the authors/editor finds them relevant. Some stylistic corrections are also suggested in the list of minor comments.**

**Minor comments:**

**Page 1 line 3: "d18O variability actually reflects…." Please refer to d18O in precipitation or mention posdepositional alterations if one refers to the d18O profiles measured in ice cores.**

Done. We changed it to "d18O variability in precipitation actually reflects …"

**Page 1 line 16: "…and the sea ice export anomaly is…" FS sea ice export anomaly in my opinion should in this study be used as an indicator/proxy of sea ice extent anomalies in the Greenland/Iceland seas. Therefore, some corrections to the text that emphasizes this point would be required.**

We agree with the reviewer. We have changed it to "… and the Fram Strait sea ice export anomaly is…". Moreover, we clarify it in Sect. 8.1 where we write "Fluctuations in this sea ice volume flux have a direct effect on the amount of open water located east and northeast of Renland."

**Page 2: Relevant for this study is a recent publication by Munch and Laepple (https://www.climpast.net/14/2053/2018/) showing the timescale dependent SNR estimates for some Antarctic ice cores.**

The reviewer is completely right and we now make a reference and comment to it in Section 5 where we defined signal to noise ratios (SNR):

".., a recent study by Munch and Laepple (2018) introduced a new way of calculating timescale-dependent SNR values which provides a basis for interpreting noise across timescales."

**Page 2 line 11 "Changes in the atmospheric circulation can be triggered by climatic oscillation modes…".Consider reformulating this statement to something like "changes in regional quasistationary modes of climate variability such as NAO can modulate (influence) global atmospheric circulation patterns"**

 We have changed the sentence to:

"Changes in regional quasi-stationary modes of climate variability such as the North Atlantic Oscillation can modulate global atmospheric circulation patterns (e.g. precipitation patterns) which affect the d18O variability."

**Page 3 line 1, ref to map Fig 1. I would suggest to add an inset with a more detailed map of the peninsula with the core locations.**

Done

**Page 3 line 5 "Inland ice", is it really needed to use a capital letter?**

It seems like both are used in the literature, but we have changed it to inland ice as we were unable to find a precise definition.

**Page 4, Caption figure 1. Change to something like "…black arrow indicates a primary direction of sea ice transport from the Arctic via the Fram Strait.**

We changed the figure and removed the black arrow (in order to also show the Greenland Sea).

**Page 5 line 15 "…where the fraction rho… ultimately is multipled…" multiplied by what? Please be specific or reformulate the sentence, it is not clear now.**

We have reformulated it to "… where the fraction rho… ultimately is multiplied onto the auxiliary diffusion length in order to transform…"

**Page 7 line 14. Adding a reference to a classical work of D. Fisher here would be highly relevant too: https://www.igsoc.org/annals/7/igs_annals_vol07_year1985_pg76-83.pdf**

We agree that this reference was notably missing and we have added it.

**Page 8 line 10 "…it is beneficial to combine the time series into a stacked record…" This was demonstrated earlier a number of times and probably can be considered trivial at this stage.**

While we agree with the reviewer that it seems rather trivial, we would prefer to keep our formulation as we want to emphasize that it is best to average records if they share a common signal. If the reviewer or editor still thinks it needs to be revised, we will change the sentence.

**Page 9 line 11-13. I would suggest to move the links to the temperature series to supplementary, if proper data citations are not available.**

We have moved the links to the reference list instead.

**Page 11 line 10 "…in said period" change to "indicated" or "studied" period?**

Changed to "studied"

**Page 11 Line 11 "decorrelation scale of this d18O-temperature relationship…". Not clear what "decorrelation scale" means in this context. Please specify.**

We changed the sentence to "…the results show that the spatial extent of this d18O--temperature relationship was different… "

**Page 12 Line 19 "The NAO describes…" Consider changing to "it can be described/approximated using the SLP difference between…". One can also improve the flow by swapping the first two sentences of section 7.**

Done. We changed the sentences to:

"A strengthening and weakening of respectively the low pressure system over Iceland and high pressure system over the Azores control both the direction and strength of westerly winds and storm tracks over the North Atlantic. Fluctuations in the difference in atmospheric pressure at sea level between Iceland and the Azores is described by the North Atlantic Oscillation (NAO)."

**Page 13 line 4 "…said correlation…" consider revising**

Changed to "… potential correlation".

**Page 13 line 6 "…multiple sea level pressure records.." or gridded datasets of SLP/GPH**

We changed it to:

"While the NAO is best described through a principal component analysis of multiple sea level pressure records or gridded datasets of sea level pressure in the North Atlantic region, …"

**Page 14 line 17. I recommend authors to refrain from referring to a specific class of ice (MYI) in the manuscript. Fram Strait ice export from the Arctic features both multi- and first year ice components. Note that some ice undoubtedly forms in situ too.**

We agree with the reviewer. We have revised it such that we now just refer to sea ice.

**Page 15 line 1. Authors are recommended to be critical regarding the quality of SIE reconstruction prior to 1900 when the original historical data density and quality gets progressively lower. It may to some extent explain lower correlations for the pre – 1900 period.**

The reviewer is right and we have added a phrasing in Sect. 8.1 where we acknowledge that it can explain some of the lower correlations.

**Page 15 line 12 "…warm temperatures result in less sea ice that can be exported away from the Arctic.. " and less ice formation locally**

Added:

"This likely indicates that warm temperatures result in less sea ice that can be exported away from the Arctic Ocean (and less sea ice formation locally)"

**Page 15 line 14 "…uncorrelated d18O-temperature relation…" consider changing to "indicates that a lack of correlation between…"**

Done.

**Page 15 line 18 "…correlation hiatus…" not sure if the use of "hiatus" is correct here.**

Changed sentence to:

"In order to examine the lacking d18O-temperature correlation, …"

**Page 15 line 20 "…only every 5 point is used…" is the analysis sensitive to the choice of the starting point for the sequence?**

This is a good point and we thank the reviewer for pointing it out.

The analysis remains the same but there are of course minor changes in the correlation coefficient calculations depending on the starting point. For a temperature anomaly above 0, the correlations will range from 0.57 to 0.87 ($p<0.05$) depending on the starting point while it will vary from -0.1 to 0.31 (all insignificant $p>0.05$) for a temperature anomaly below 0. We choose to keep our initial starting point in order to include an extra data point in the analysis (35 instead of 34). While we present a correlation coefficient of 0.83 (which isn't the highest value), the exact correlation coefficient is not important for the conclusion of this work as we only want to demonstrate that the d18O-temperature relation solely exists during warm years (which it does independently of starting point).

**Page 15 line 24 "…less multiyear sea ice…" see my earlier comment**

We removed "multiyear" from the sentence.

**Page 15 line 31. In terms of sea ice variability, May is considered largely a winter month**

We agree with the reviewer and we have accounted for this in the analysis by shifting the seasonal averaging of sea ice extent 1 month (summer: June-November; winter: December-May). This modification made the correlations increase a bit for the winter and summer averages but the overall results remain identical to before.

**Page 19 line 5: "…precipitation has journeyed further…" better refer to a longer distillation path/trajectory**

Done

**Page 19 line 22 "correlations stopped in the…." "diminishes"?**

Changed.

**Page 20 line 2 "…a vapour mass experiences (en route) from…"**

Added.

**Reviewer 2**

**The authors demonstrate new high-resolution stable water isotope records from Renland peninsula (East Greenland) and make an attempt to establish the relationship between the isotopic composition and the regional instrumental air temperature data. They show that the temperature in the condensation site is not the only driver, the other factors like sea ice concentration in the nearby ocean and possibly regional circulation pattern (NAO) being important. Thus the complex nature of the isotopic composition of the atmospheric precipitation is clearly demonstrated. The obtained results are important for the interpretation of the isotopic content of the fossil precipitation retrieved as firn and ice cores by means of deep drilling of the polar ice sheets.**

**To my opinion, the manuscript can be published with minor revisions, as listed below:**

**Page 1, lines 6-7: consider rewording, e.g. "and examine which amount of this variability could be attributed..."**

We changed it to: "The objective of this study is therefore to evaluate the d18O variability of ice cores drilled on Renland and examine what amount of the signal that can be attributed to regional temperature variations"

**Page 2, line 2: from the context of this paragraph it's clear that you are talking about polar ice sheets, so it's better to write "polar ice sheets" instead of "ice caps" here.**

Done

**Page 3, line 3 - the resolution of the map in Figure 1 does not really allow to see it.**

We have made an enhanced figure where the site locations should be better visualized.

**Page 3, line 5 and below - Inland or inland?**

We changed it to inland

**Page 3, line 14 - "as we here have" sounds awkward for me.**

We changed it to "…the study focuses on the period AD 1801-2014 where three overlapping ice core records and instrumental temperature recordings are available."

**Page 9, line 2 - in order to filter out**

Done

**Page 14, line 7 - unwanted comma in the end of the sentence**

Thanks for noticing that. It has been fixed.

**Page 14, line 11 - should be 1821-1909?**

Correct! Thank you for noticing that. It has been fixed.

**Page 15, line 21, and Figure 11 - the term "normalized" is usually defined by (value - mean)/standard_deviation. What is shown here is a temperature anomaly (= deviation from mean).**

This has been changed to anomaly in both Figure 11 and in the text

**Short comment**

**Dear Authors,**

**Congratulations to the interesting study. I've two points for this short comment.**

**1, Can you make available the raw d18O series of the Renland ice cores? or the annualized d18O records? I cannot find any link either to a data repository or to a supplementary source where the ice core derived d18O data could be available.**

We will make the annualized d18O data available on PANGAEA and and on http://www.iceandclimate.nbi.ku.dk/data/ upon publication. This is now specified in the manuscript.

**2, Changes in the seasonal distribution of precipitation could be also a potential explanation for temporal changes is d18O-temperature relation. Even seasonal distribution of precipitation since 1910 might explain the good correlation with annual mean air temperature. However, an increased seasonal contrast in the intraanual distribution of precipitation before 1910 might introduce a seasonal bias to the depositional d18O record and it can disturb the correlation with the annual mean air temperature. It can be tested if amount-weigthed mean temperatures could be calculated and involved also into the correlation analysis. A recent study from Polish Polar Research (https://content.sciendo.com/view/journals/popore/38/2/article-p105.xml) can be potentially considered in this idea.**

**best regards, Zoltan Kern**

We agree that changes in the seasonal distribution of precipitation could explain some of changes in the d18O-temperature relation. We have therefore added the following comment in the discussion

"Alternatively, if the seasonal distribution of precipitation on Renland changed significantly prior to the 1910s, it could lead to a change in the relationship between the d18O signals and temperature.

While the dating resolution does not permit a direct assessment of such changes, we do observe that the difference between the summer and winter d18O have in fact changed over the 1801-2014 period (Fig. 3)."

However, it is unfortunately not possible to implement the amount-weighted mean temperatures as we do not have any monthly (or seasonal) instrumental precipitation or accumulation rate records available for the Renland drill sites.

**References**

Steffen, K. and Box, J. (2001). Surface climatology of the Greenland ice sheet: Greenland Climate Network 1995-1999.